# Adaptive Optics Imaging to Analyze the Photoreceptor Layer Reconstitution in Acute Syphilitic Posterior Placoid Chorioretinopathy

**DOI:** 10.3390/life12091361

**Published:** 2022-08-31

**Authors:** Fabrizio Giansanti, Stefano Mercuri, Lorenzo Vannozzi, Andrea Govetto, Angelo Maria Minnella, Tomaso Caporossi, Alfonso Savastano, Maria Cristina Savastano, Gloria Gambini, Stanislao Rizzo, Gianni Virgili, Daniela Bacherini

**Affiliations:** 1Department of Neurosciences, Psychology, Drug Research, and Child Health, Eye Clinic, University of Florence, AOU Careggi, 50139 Florence, Italy; 2Ophthalmology Department, Asst-Settelaghi, University of Insubria, 21100 Varese, Italy; 3Ophthalmology Unit, Fondazione Policlinico Universitario A. Gemelli IRCSS, 00168 Rome, Italy; 4Ophthalmology Unit, Catholic University “Sacro Cuore”, 00168 Rome, Italy; 5Consiglio Nazionale delle Ricerche, Istituto di Neuroscienze, 56127 Pisa, Italy

**Keywords:** acute syphilitic posterior placoid chorioretinopathy, adaptive optics, optical coherence tomography, optical coherence tomography angiography, syphilis, uveitis

## Abstract

Acute posterior syphilitic placoid chorioretinopathy (ASPPC) is a rare ocular manifestation of syphilis characterized by outer retinal layers involvement and drop in visual acuity. The current work documents outer retinal layer involvement in this pathology and their reconstitution with treatment by means of adaptive optics (AO). Three eyes of two patients together with four controls eyes were included in the study. Patients underwent optical coherence tomography (OCT) and OCT angiography (OCTA) scan centered on fovea, where vessel density (VD) and vessel perfusion (VP) were calculated. AO images centered on fovea were acquired and cone density (CD) and cone spacing (CS) were measured and compared to control group. Multimodal imaging was performed at presentation, at 10 days, and at 2-month follow-up. All eyes improved in visual acuity, with reconstitution in outer retinal layers at 2-month follow-up. Overall choriocapillary layer VD and VP improved. AO imaging was able to identify outer retinal alterations at presentation and at follow-ups, with improvement in tissue architecture. CD and CS was respectively lower and greater than controls at all follow-ups and improved within patients at the 2-month follow-up. In conclusion, AO was able to document outer retinal alterations in ASPPC at presentation and improvement over the follow-up, representing a tool to study photoreceptor layer involvement in this pathology.

## 1. Introduction

Acute syphilitic posterior placoid chorioretinopathy (ASPPC) is a rare ocular manifestation in the clinical spectrum of syphilis, characterized by one or multiple yellowish placoid lesions at the posterior pole, associated with disruption of the outer retina, alteration of the ellipsoid zone (EZ) and with variable expression of vitreous and choriocapillaris (CC) inflammation [1,2]. Decrease in visual acuity is dependent on involvement of the foveal area, as patients are typically young and without other comorbidities. If treated promptly, clinical response in these patients is generally fast and visual recovery is excellent, as also documented by reconstitution of the EZ after treatment, as described by Pichi et al. [3].

Flood-illumination adaptive optics (AO) is a novel medical device able to image with high-resolution retinal microstructures, including photoreceptors [4]. Although this pathology is rare to encounter in the general practice, clinical studies have documented the different phases of the disease from the morphological and angiographical perspective [3,5,6,7,8]. The role of different pathogenetic components represent a matter of debate as the clinical manifestation may be the resulting expression of direct tissue infection, reactive inflammation, or a combination of the two [9,10,11,12].

In the present study, our aim is to describe AO imaging findings of outer retina and photoreceptor layer involvement at baseline and to study their structural characteristics during reconstitution after therapy in two patients affected by ASPPC, integrating results from multimodal imaging.

## 2. Materials and Methods

Study design was a longitudinal case series. Study protocol was carried out at Careggi Teaching Hospital in Florence, Italy. This research adhered to the tenets of the Declaration of Helsinki with written informed consent obtained from all patients.

Patients with diagnosis of ASPPC were enrolled in the study. Exclusion criteria were presence of any other retinal disorder potentially threatening retinal architecture, optical media opacities which could alter image analysis. Diagnosis of ASPPC was based on positive serology for syphilis and on clinical characteristics at fundus biomicroscopy, spectral domain optical coherence tomography (SD-OCT), fundus auto-fluorescence (FAF) and fundus fluorescein angiography (FFA) [3]. Treponemal (treponema pallidum hemoagglutination assay—TPHA) and non-treponemal tests (rapid plasma reagin—RPR) were tested in study subjects together with HIV serology screening and complete blood count. Baseline clinical and anatomical measurements were collected at time of patient presentation and further follow-up visits (10 days after therapy, 2 months after therapy initiation). Patients underwent complete ophthalmological examination: measurement of best-corrected visual acuity (BCVA) using Snellen chart, intraocular pressure (IOP) evaluation by means of Goldmann applanation tonometry, fundoscopy, SD-OCT, and optical coherence tomography angiography (OCTA) performed by means of Spectralis HRA-OCT (Heidelberg Engineering; Heidelberg, Germany) and NIDEK (RS-3000 Advance 2 spectral domain OCT; NIDEK Co. Ltd., Gamagori, Japan). Adaptive optics imaging was acquired using a flood-illuminated AO retinal camera (rtx1 TM adaptive optics (AO) retinal camera Imagine Eyes, Orsay, France) [13].

All the imaging acquisitions were obtained at the foveal level and all recruited eyes had involvement of the fovea and analyzed areas by the placoid lesions.

### 2.1. Spectral-Domain Optical Coherence Tomography and Fundus Autofluorescence

The acquisition protocol included SD-OCT horizontal scans, passing through foveal center. Structural B-scans were independently analyzed by two examiners (SM, DB). SD-OCT scans were performed to state disease activity, integrity of external limiting membrane (ELM) and ellipsoid zone (EZ), integrity of RPE layer, status of choriocapillaris (CC). Status of ELM and EZ could be defined as “present”, “disrupted”, or “absent”. Integrity of RPE could be categorized as “present” or “damaged”.

Appearance at FAF at every follow-up was described based on increased (hyper-autofluorescence) or decreased autofluorescence (hypo-autofluorescence) and qualitative description of RPE damage.

### 2.2. Optical Coherence Tomography Angiography

OCTA analysis included 3 mm × 3 mm acquisitions, centered on the fovea. The superficial capillary plexus (SCP), deep capillary plexus (DCP), and choriocapillaris (CC) were segmented in the macula. Each automated segmentation was carefully reviewed and manually corrected, if necessary, by an expert ophthalmologist (DB). The default RS-3000 Advance 2 OCTA AngioScan software was used to evaluate the vessel density (VD) (ETDRS-based vessel density (%)) SCP (SCP VD) and DCP (DCP VD) and the vessel perfusion (ETDRS-based vessel perfusion (%)) of the SCP (SCP VP) and the DCP (DCP VP), according to the nine ETDRS subfields. Vessel density (VD) was defined as the absolute length of the perfused vasculature per area in a considered region of estimation. Its units are mm/mm^2^ ranging from 0 (no vessels) to an unbounded value. Vessel perfusion (VP) is characterized as the complete zone of perfused vasculature per unit of the considered area. It is expressed as a percentage extending from 0% (no perfusion) to 100% (completely perfused). Vessel and perfusion densities of the SCP, DCP, and CC were automatically calculated by the software on OCTA 3 mm × 3 mm volume scans in the whole foveal and the inner and outer retina. Using OCTA, we assessed the presence of flow voids in the CC layers, defined as areas of decreased or absent perfusion in the CC vascular architecture [14].

### 2.3. Adaptive Optics Imaging

Along with AO imaging an 850 nm LED was used to illuminate the fundus of the eye and a low noise CCD camera was used to capture the raw images of the retina at 10 Hz over a 4 × 4 field of view. Refractive error and axial length were required by the device for correction of focus to directly image cones layer and assess cone photoreceptor counts in all eyes. Focus was furtherly manually adjusted to achieve the best focus on photoreceptors plane.

Cone density was analyzed in a region of interest (ROI) in the affected retina. This was carried out using a custom software package AO Detect Mosaic V.0.1 (Orsay, France), Imagine Eyes, Orsay, France. The manufacturer’s software automatically detects the cone mosaic and the position of the photoreceptors, and this allows us to quantitatively evaluate the photoreceptors. The software detects hyper-reflective small circular spots with different reflectivity from the surrounding background; the spatial distribution of the centers of these spots was analyzed in terms of local cell numerical density (number of cones per square millimeter of retinal surface; cone/mm^2^) [15,16,17,18,19,20]. Overall cone density and overall cone spacing was found, including calculations from eight ROI in every patient, similar to work previously done by Forte et al. [21].

Cone analysis values were grouped according to region at the macular level at 1.5° from the foveal center (superior, superonasal, nasal, inferonasal, inferior, inferotemporal, temporal, and superotemporal to fovea and overall cone density and spacing including all values) and results were gathered at each follow-up. Cone density and CS values were obtained also from a group of age, sex, and refractive error-matched controls and compared to patients’ group.

### 2.4. Statistical Analysis

The continuous variables were presented with mean ± standard deviation (STD); analysis with differences between baseline and follow-up parameters, as well as the difference between treatment groups was performed by means of SPSS (Version 22, IBM Corp, Armonk, NY, USA).

## 3. Results

Three eyes of two patients (two males, aged 44 and 50, mean 47.0 years old) with diagnosis of acute syphilitic posterior placoid chorioretinitis (ASPPC) were enrolled in the study. Both subjects had no relevant past ophthalmologic and general medical history, with no medications taken prior to first ophthalmological examination. They both presented to our clinic with a fast-progressing drop in visual acuity (VA), with Patient 1 referring symptoms to both eyes and Patient 2 just in the left eye. The fellow eye of Patient 2 did not show any manifestation of the disease or decrease in VA. Diagnosis of infection from syphilis was confirmed in both patients by positive serologic tests (TPHA and RPR), serology for HIV was negative in both subjects and complete blood count was within normal limits. Patients were admitted in the infectious diseases department, where underwent lumbar puncture to confirm diagnosis of neurosyphilis. Magnetic resonance imaging was performed to rule out presence of central nervous systems lesions secondary to syphilis infection, with none of these lesions detected. Intravenous antimicrobial therapy with 12 million units of Penicillin per day were administered for 10 days. A clinical and serological resolution was confirmed after 1 month from the beginning of the antibiotic therapy in both patients, with reconstitution in outer retinal architecture and improvement in outer retinal features characteristic of the pathology.

In Figure 1 improvement of clinical features of the disease is depicted from presentation over the follow-up (10 days, 2 months) at the fundoscopic level, at OCT and FAF imaging. Contours of the placoid lesion at FAF and fundoscopy became more defined in the first days after therapy initiation, and slowly faded over the follow-up together with normalization of autofluorescence. External limiting membrane recovered fully in all three eyes; EZ progressively but fully recovered in two eyes with partial reconstitution in one eye; minimal RPE alterations were present in all eyes at 2-month follow-up.

### 3.1. Optical Coherence Tomography

Clinical data regarding BCVA and outer retinal layers status are depicted in Table 1. Visual acuity significantly recovered, reaching 0.00 LogMAR (20/20 Snellen) in two eyes and 0.10 LogMAR (20/25 Snellen) in one eye at the 2 months follow up. Outer retinal layers improved in morphology over the follow-up, with ELM recovering fully in all three eyes; EZ fully recovered in two eyes and partially reconstituted in one eye; minimal RPE alterations were present in all eyes at 2-month follow-up; RPE–Bruch complex morphology recovered partially in two eyes (66%) and completely in one eye (33%).

Best corrected visual acuity (BCVA), expressed in LogMAR, optical coherence to-mography (OCT) and fundus autofluorescence (FAF) characteristics of ASPPC patients over the follow up. Outer retinal characteristics analyzed at OCT include integrity of ex-ternal limiting membrane (ELM), ellipsoid zone (EZ) and retinal pigmented epithelium (RPE). All eyes improved over the follow up with outer retinal layers reconstitution and minimal residual RPE damage.

### 3.2. Optical Coherence Tomography Angiography

OCT-A derived VD and VP are summarized in Table 2. Overall VD and VP at the macular level in the SCP, DCP and CC improve over the follow-up, and areas of choriocapillaris vessel flow voids are detectable at presentation and decrease over the follow up, as shown in Figure 2.

### 3.3. Adaptive Optics Imaging

Adaptive optics imaging of patients over the follow-up and AO imaging of controls is shown in Figure 3. Four eyes of four age, sex, and refractive error-matched subjects control group were recruited (four males, mean age 47.5 ± 2.0) for image analysis, cone density (CD), and cone spacing (CS) comparison.

At presentation, AO images at the foveal level showed damage on the photoreceptor layer–RPE complex, with interruptions in the tissue architecture and appearance of dark-patchy areas, not detectable in controls. Together with improvement in visual acuity and outer retinal layers reconstitution at SD-OCT, we documented tissue architecture improvement and a reduction in extension and number of these dark patchy areas. Although improvement in overall architecture, tissue disruption still persisted at the 2-month follow-up compared to control eyes.

Data from AO imaging regarding CD and CS of patients at each follow-up and controls are depicted in Table 3.

Mean cone density at 1.5° from the fovea in the ASPPC group was lower than in the control group both at presentation and at 2-month follow-up, whereas cone spacing was greater. After the clinical resolution of the placoid lesion, overall photoreceptor density increased in the follow-up in each eye, whereas cone spacing decreased. The CD and CS values of the unaffected eye of Patient 2 are also displayed and were not different from control subjects, as seen in Figure 4.

Comparing OCTA 3 mm × 3 mm images of CC layer at the foveal level with AO imaging, we observed a correspondence between hypo-perfused areas in the CC layer and these dark patchy areas with a low density of photoreceptors on AO, as shown in Figure 5. Areas of decreased VD at presentation improved during the follow-up and an improvement with tissue reconstitution corresponded at AO imaging in these areas.

## 4. Discussion

Acute syphilitic posterior placoid chorioretinopathy (ASPPC) represents a rare manifestation of posterior uveitis in patients infected by treponema pallidum [1,2,3]. This clinical entity belongs to the class of placoid lesions that includes serpiginous choroiditis or multiple evanescent white dot syndrome (MEWDS), which are diseases with an immunological-based pathogenesis [10].

This opens a debate about the possible pathogenesis of ASPPC, as the clinical phenotype may derive from direct infection of the tissue, from inflammation subsequent to the infection, or a combination of the two, generating an occlusive/thrombotic insult to the CC layer, which then leads to damage to overlying photoreceptors [5,6,7,8,9]. Hyper-reflective outer retinal lesions represent the hallmark of ASPPC, together with ellipsoid zone (EZ) alterations at structural optical coherence tomography (OCT) scan. Resolution of placoid lesion with antimicrobial therapy leads in most cases to partial or complete reconstitution of outer retinal layers and normalization at fundus autofluorescence (FAF). Hyper-autofluorescence seen in ASPPC patients as well as in our cohort arises from damage at the level of the outer retinal complex. It may derive from a combination of hypermetabolism of damaged RPE layer and lack of masking effect of rhodopsin, which decreases in density with damage to photoreceptor outer segments, as seen in other diseases affecting photoreceptors [22]. Multimodal imaging is an essential approach for studying retinal diseases, allowing evaluation of retinal morphological characteristics while assessing inner and outer retinal layer functional status and tissue vascular perfusion. Optical coherence tomography angiography (OCTA) is as relatively new non-invasive technique able to assess microvascular perfusion qualitatively and quantitatively in a region of interest and able to distinguish retinal and choroidal plexi and their involvement in ocular diseases. Microvascular choriocapillary layer (CC) involvement in ASPPC has been described by Barikian et al., where CC flow deficits corresponded to placoid lesions at fundoscopy and flow improvement followed its resolution after therapy [8].

In our study, we also showed a decrease in vessel density in the CC layer in affected lesions, with improvement in flow voids areas at 2 months follow-up.

Adaptive optics (AO) imaging represents a relatively novel technique to morphologically assess different retinal structures, including retinal vessels, vitreoretinal interface, and photoreceptors and outer retinal status [4,5,6,7,8,9,10,11,12,13].

To date, this is the first study that documents clinical features using AO imaging of the outer retinal layers in ASPPC. We documented by means of AO overall decrease in density of photoreceptors together with tissue architecture disruption, with patchy dark areas devoid of photoreceptors in the areas affected by the placoid lesion. The disease causes alterations at the retino-choroidal interface, well detectable using structural OCT, enface OCT slab, and FAF, which improve in morphology at 2 months follow-up together with improvement in visual acuity [3]. In our cohort of patients, follow-up from AO imaging after 2 months from therapy initiation showed an improvement in tissue architecture in accordance with reconstitution of outer retinal layers at SD-OCT and visual acuity gain. At this follow-up, although the overall improvement in tissue architecture, AO highlighted that tissue remains more disorganized compared to controls. We believe that degree of disarrangement at AO may be dependent on the damage of the RPE layer after therapy completion, which may be proportional to intensity of manifestation and duration of the disease prior to treatment initiation with antimicrobial therapy.

The right eye of Patient 1 depicted in Figure 1, which suffered a profound disruption of outer retinal layers at presentation, together with a strong drop in visual acuity, showed a present EZ with only partial disruption at SD-OCT at 2-month follow-up, together with a persistent disorganization of photoreceptors at AO. The left eye of patient 2, depicted in Figure 4 and which had a milder form of ASPPC, disclosed only focal AO alterations at 2-month follow-up. AO, able to image a narrower region of interest with higher resolution, reveals with higher accuracy outer retinal tissue damage. AO may thus represent a useful tool to study outer retinal diseases, characterized by disarrangement of the photoreceptor–RPE–Bruch’s membrane interface, and to detect subclinical outer retinal alterations at the photoreceptor or RPE level.

We then found a correspondence between areas of rarefaction of the CC layer detected using OCTA correspond to areas of higher alteration of photoreceptors and RPE–Bruch complex at AO imaging. In the same fashion, improvement in CC VD at OCTA corresponded to decrease in size and number of dark patchy areas representing areas devoid of photoreceptors.

We are aware that our study has several limitations, among all the scant number of eyes imaged and a short follow-up. This pathology is a rarity in the clinical practice and AO is a technology which has still limited use in routine practice, as its application in the multimodal imaging is limited to few specialized centers. Improvement in the number of patients recruited and increase in follow-up length would improve disease characterization and our understanding of the disease.

AO is a relatively novel technique and allows a direct visualization of photoreceptor outer segments in healthy and pathologic eyes [15,16,17,18,19,20]. This technique carries some limitations, as AO analysis is not fully automated because the area of analysis for cone count was selected manually, as previously done [21]. To overcome this limitation, a control group of healthy patients evaluated with the same AO modalities was used to assess the variation in tissue architecture and cone photoreceptor density in ASPPC, as seen in Figure 3. Furthermore, depth of focus to image photoreceptors needs occasionally to be corrected manually, despite an accurate adjustment of focus through axial length and refractive error correction by the instrument. All eyes analyzed had axial length within normal limits, with emmetropic refraction, and image acquisition was consistently repeatable with high image definition. We are also aware of the difficulty in interpretation and analysis of images by the software in diseased eyes, due to architecture disruption. OCTA techniques are known to display artifacts, and evaluation of the CC may be altered by blockage of light by the RPE [23].

Another important drawback is the lack of a histopathological correlation able to confirm our findings of decrease in number and density of photoreceptors outer segments.

This study represents a first investigation regarding the role of AO imaging in the assessment of outer retinal involvement in ASPPC and may represent one of the first approaches for quantification of photoreceptors in these pathologies. Further prospective studies are needed to confirm our findings. In ASPPC, with the aid of AO imaging, we can assess with higher resolution the tissue morphology and to detect microscopic changes that may be undetected with OCT.

## Figures and Tables

**Figure 1 life-12-01361-f001:**
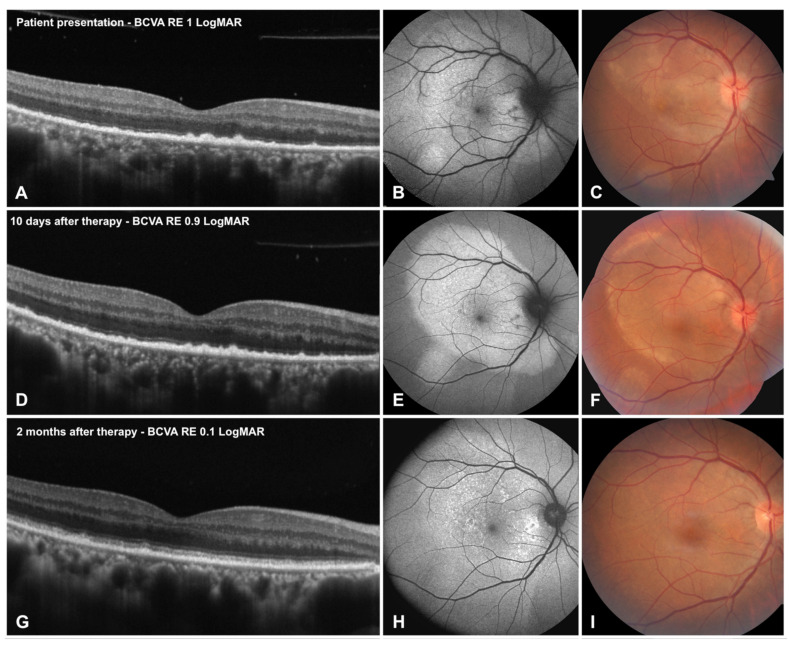
Improvement in the right eye of Patient 1 at optical coherence tomography (OCT), fundus autofluorescence (FAF) and fundoscopy from presentation (**A**–**C**), to 10 days follow-up (**D**–**F**) to 2 months after treatment (**G**–**I**). Borders at the placoid lesion become more defined and slowly fade with decreasing hyper-autofluorescence. In this eye external limiting membrane and ellipsoid zone which were disrupted at presentation, recovered their morphology, together with improvement at FAF.

**Figure 2 life-12-01361-f002:**
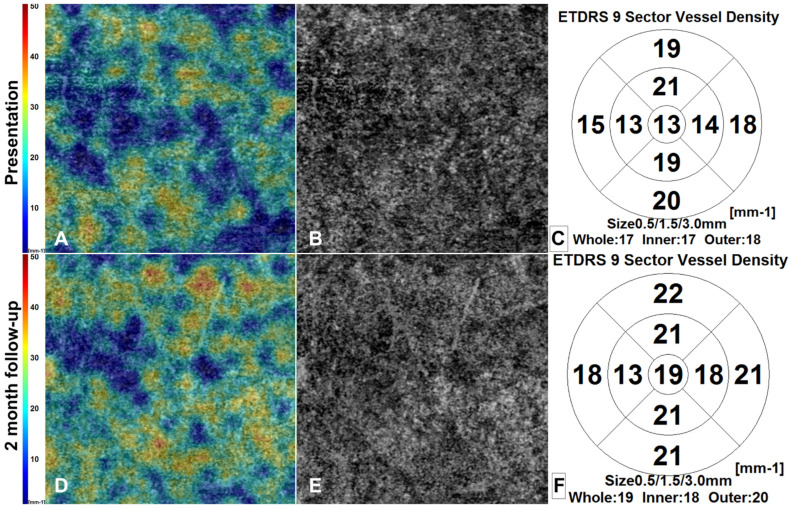
3 × 3 mm optical coherence tomography angiography (OCTA) choriocapillary layer density map (**A**,**D**), image reconstruction (**B**,**E**), and ETDRS chart density map (**C**,**F**) of right eye of Patient 1. Over the 2-month follow-up we see an improvement of flow voids identifiable at presentation.

**Figure 3 life-12-01361-f003:**
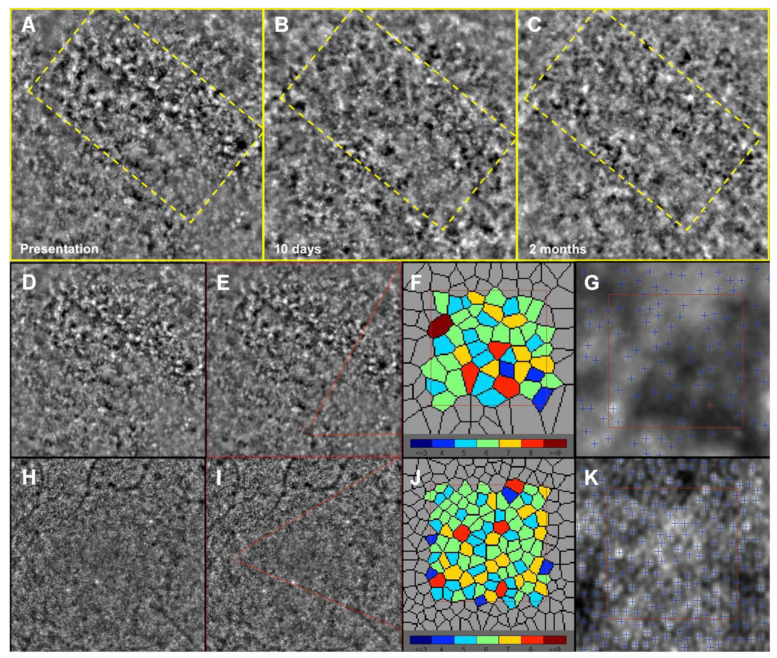
Progressive improvement in tissue architecture with adaptive optics. The top 3 yellow squares depict the right eye of Patient 1 at time of patient presentation (**A**), and along follow-up at 10 days (**B**) and 2 months (**C**). At presentation, defined dark patchy areas were present, representing outer retinal tissue disorganization (yellow dashed rectangle), while these areas decrease in extension and number, with partial tissue architecture restoration. It also shows adaptive optics Imaging analysis in patients eyes (**D**–**G**) and controls (**H**–**K**). Cone density (CD) and spacing (CS) analysis were measured in regions of interest at 1.5 degrees from foveal center (red square), and individual cones could be quantitatively estimated.

**Figure 4 life-12-01361-f004:**
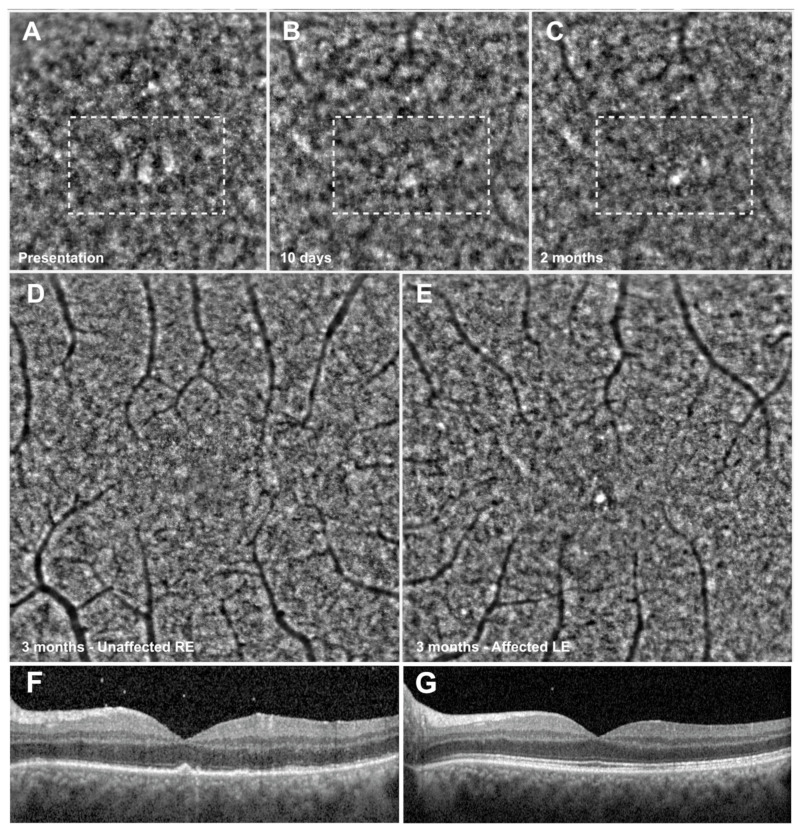
Tissue architecture improvement with adaptive optics. The top 3 squares depict the left eye of Patient 2 at time of patient presentation (**A**), and at follow-up of 10 days (**B**) and 2 months (**C**). At presentation, tissue disorganization (dashed rectangle) with architecture restoration over the follow-up was shown. Images also show the difference between the unaffected right eye of Patient 2 (**D**) compared to the affected left eye (**E**) 3 months after therapy initiation. Although visual acuity recovered completely, the tissue remained partially disorganized, with slight presence of dark patchy areas not present in the fellow eye. The bottom images (**F**,**G**) show the affected left eye at presentation and at 3-month follow-up. At 3 months (**G**), there is complete reconstitution of outer retinal layers, with only a slight granular appearance of ellipsoid zone, which corresponds to damage seen at adaptive optics.

**Figure 5 life-12-01361-f005:**
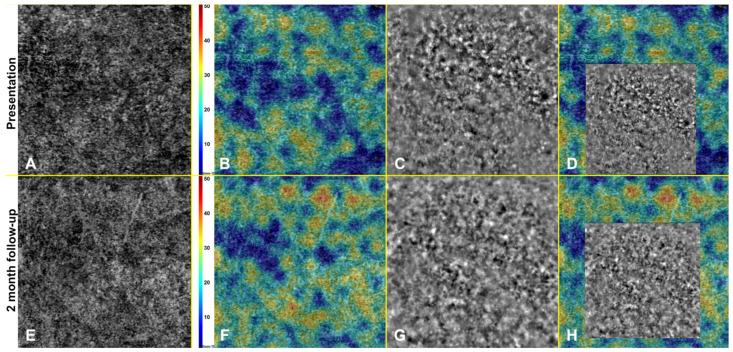
Highlights the correspondence between 3 mm × 3 mm optical coherence tomography angiography (OCTA) choriocapillary (CC) images (**A**,**E**) and OCTA vessel density map (**B**,**F**) with adaptive optics images at the foveal level (**C**,**G**). Images (**A**–**D**) refer to patient at presentation, while images (**E**–**H**) at 2-monthsfollow-up. Tissue architecture disruption corresponds to area of decreased vessel density (**D**) at OCTA, while tissue partial recovery at 2-month follow-up corresponds to decrease in flow-void areas at the CC level (**H**).

**Table 1 life-12-01361-t001:** Summarizes best corrected visual acuity (BCVA), expressed in LogMAR, op-tical coherence tomography (OCT), and fundus autofluorescence (FAF) characteristics of ASPPC patients over the follow up. Outer retinal characteristics analyzed at OCT include integrity of external limiting membrane (ELM), ellipsoid zone (EZ) and retinal pigmented epithelium (RPE). All eyes improved over the follow up with outer retinal layers recon-stitution and minimal residual RPE damage.

Patient 1 RE	Presentation	10 Days	2 Months
**BCVA**	1.00	0.9	0.1
**ELM**	Disrupted	Present	Present
**EZ**	Absent	Disrupted	Present with focal damage
**RPE**	RPE clumping	RPE clumping	Disrupted
**FAF**	Hyper-autofluorescence with RPE alterations and blurred margins	Hyper-autofluorescence with RPE alterations with defined margins	Attenuated diffused hyper-autofluorescence with RPE alterations
**Patient 1 LE**			
**BCVA**	0.6	0.3	0.0
**ELM**	absent	disrupted	present
**EZ**	absent	absent	present
**RPE**	RPE clumping	disrupted	disrupted
**FAF**	Hyper-autofluorescence with mild RPE alterations	Hyper-autofluorescence with mild RPE alterations	minimal RPE alterations
**Patient 2 LE**			
**BCVA**	0.3	0.2	0.0
**ELM**	disrupted	present	present
**EZ**	disrupted	disrupted	present
**RPE**	RPE clumping	RPE clumping	present
**FAF**	Hyper-autofluorescence with minimal RPE alterations	Hyper-autofluorescence with minimal RPE alterations	minimal RPE alterations

**Table 2 life-12-01361-t002:** Highlights values of vessel density (VD) and vessel perfusion (VP) at optical coherence tomography angiography at the level of superficial (SCP), deep and choriocapillary (CC) vascular plexi. Mean values (mVD/mVP) and standard deviation are displayed, resulting from mean value of all ETDRS sector in the retina, comparing tissue vascular supply at presentation (t0) and at 2 months after treatment (t1). There is an overall increase in VD and VP over the follow up.

OCTA VD	t0	t1	OCTA VP	t0	t1
**SCP mVD**	15 ± 0.2	16.5 ± 1.4	**SCP mVP**	45.4 ± 4.3	47.9 ± 2.0
**DCP mVD**	11.8 ± 3.9	15.3 ± 1.3	**DCP mVP**	18.9 ± 1.4	28.8 ± 10.9
**CC mVD**	18.7 ± 1.9	20.8 ± 1.8	**CC mVP**	48 ± 2.2	55.5 ± 2.7

**Table 3 life-12-01361-t003:** Depicts cone density (CD) and cone spacing (CS) values of patients at presentation (cone density/spacing 1) and at 2 months follow-up (cone density/spacing 2), as well as controls, expressed as mean and standard deviation (STD). There is an overall increase in cone density at the macular level in patients over the follow-up, but cone density remains significantly lower, while cone dispersion higher compared to controls. Values of CD and CS in unaffected eye of Patient 2 are displayed, which are comparable to control subjects.

	Cone Density 1	Cone Density 2	Cone Spacing 1	Cone Spacing 2
**1.5° from Fovea**	**Mean ± STD**	**Mean ± STD**	**Mean ± STD**	**Mean ± STD**
**superior**	13,418 ± 3309	16,804.6 ± 2356	9.49 ± 0.95	8.59 ± 0.69
**superonasal**	12,004.1 ± 2802	**15,699 ± 2405**	9.57 ± 1.01	8.66 ± 0.70
**nasal**	11,661.6 ± 2933	15,403.3 ± 3307	10.16 ± 1.21	8.92 ± 0.83
**inferonasal**	12,302 ± 2560	**15,101 ± 2702**	9.78 ± 0.91	8.96 ± 0.52
**inferior**	12,904.6 ± 2387	14,879.3 ± 377	9.69 ± 0.78	9.01 ± 0.17
**inferotemporal**	12,420 ± 2101	15,201 ± 2100	9.81 ± 0.95	9.00 ± 0.56
**temporal**	11,966.6 ± 2489	15,559.6 ± 725	9.91 ± 0.86	8.81 ± 0.14
**superotemporal**	12,231 ± 2305	15,702 ± 1607	9.82 ± 0.88	8.77 ± 0.23
**overall**	12,364 ± 2610	15,543 ± 1947	9.77 ± 0.94	8.84 ± 0.48
	**Controls Cone Density**		**Controls Cone Spacing**	
**1.5° from Fovea**	**Mean ± STD**		**Mean ± STD**	
**superior**	26,580 ± 3380		6.91 ± 0.43	
**superonasal**	25,760 ± 2102		6.99 ± 0.34	
**nasal**	24,293 ± 1369		7.08 ± 0.19	
**inferonasal**	23,990 ± 1202		7.01 ± 0.10	
**inferior**	24,353 ± 1154		7.06 ± 0.14	
**inferotemporal**	23,902 ± 2001		7.26 ± 0.3	
**temporal**	22,514 ± 2423		7.43 ± 0.43	
**superotemporal**	24,002 ± 2100		7.22 ± 0.26	
**overall**	24,424 ± 1966		7.12 ± 0.27	
	**Unaffected Eye of Patient 2**		**Unaffected Eye of Patient 2**	
**1.5° from Fovea**	**Values**		**Values**	
**superior**	24,580		7.02	
**superonasal**	24,230		7.12	
**nasal**	23,819		7.22	
**inferonasal**	23,902		7.20	
**inferior**	23,410		7.13	
**inferotemporal**	23,450		7.35	
**temporal**	22,623		7.49	
**superotemporal**	22,995		7.40	
**overall**	23,626		7.24	

## Data Availability

Not applicable.

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
