# Peer review of "Adaptive Optics Imaging to Analyze the Photoreceptor Layer Reconstitution in Acute Syphilitic Posterior Placoid Chorioretinopathy"

_life, 2022, doi:10.3390/life12091361_

Round 1

Reviewer 1 Report

please open the attached file.

Author Response

Dear Editor and Reviewer, we are grateful for the opportunity of submitting a revised version of our manuscript. We have corrected and implemented the text, following the useful suggestions of the Reviewers. We hope that our efforts can be positively considered.

See the attached file for Reply to Reviewer 1

Reviewer 2 Report

Giansanti and colleagues provide a really nice depiction of the longitudinal progression of Acute Syphilitic Posterior Placoid Chorioretinopathy. The presentation and figures are of high quality and the idea of using adaptive optics to image photoreceptors in such lesions is sufficiently novel to warrant publication.

My first and major concern is the small sample size (3 eyes) which I understand is justifiable by the rare incidence of the disorder. However, such a small sample does not support the comparative statistical testing done by the authors.

My second concern is the possible misinterpretation of photoreceptor density owing to the segmentation capacity of the AO machine. In early involvement, placoid lesions protrude inwards into the outer retinal tissue, which could result in inward pushing of the outer retinal layers (as obvious in figure 4). Since it is assumed that the AO acquires a single flat focus of the photoreceptor layer, the devoid areas may be due to even segmentation rather than actual disruption.

Author Response

Dear Editor and Reviewer,
We are grateful for the opportunity of submitting a revised version of our manuscript.
We have corrected and implemented the text, following the useful suggestions of the Reviewers.
We hope that our efforts can be positively considered.

Sincerely yours,

Stefano Mercuri

Please find comments in the attached file

Reviewer 3 Report

The authors used AOSLO imaging to analyze patients with acute syphilitic posterior placoid chorioretinopathy. The disease is rare and the approach is novel. However, the most critical flaw of the study is that they analyzed only two patients. The result would be more suitable to case report journals. Other comments are below.

1.       The authors showed decreased cone density in patients at presentation, and it partially recovered after two months. However, it is believed that cone cells do not increase its number. The measurement certainly reflect some changes, but the device cannot differentiate the absence and inability to image cone cells. Actually, Figure 3 panel G is blurred indicating that imaging an disease eye is challenging.

2.       Cone density and spacing was calculated from 4 ROI in each eye. It is OK for normal eyes and some dystrophy with homogenous degeneration. But in the investigated disease, the result may be totally different depending on the investigated area. Anyway, the result is from two patients and statistical analysis is not very relevant to the study design.

Author Response

We are grateful for the opportunity of submitting a revised version of our manuscript.

We have corrected and implemented the text, following the useful suggestions of the Reviewers.

We hope that our efforts can be positively considered.

Sincerely yours,

Stefano Mercuri

Please find comment in the attached file

Round 2

Reviewer 3 Report

The authors improved the manuscript significantly. The reviewer has no further comment, but still think the paper is suitable to more specialized journals.